# SemGIR: Semantic-Guided Image Regeneration based method for AI-generated Image Detection and Attribution

## ABSTRACT

The rapid development of image generative models has lowered the threshold for image creation but also raised security concerns related to the propagation of false information, urgently necessitating the development of detection technologies for AI-generated images. Presently, text-to-image generation stands as the predominant approach to image generation, where the rendering of generated images hinges on two primary factors: text prompts and the inherent characteristics of the model. However, the variety of semantic text prompts yields diverse generated images, posing significant challenges to existing detection methodologies that rely solely on learning from image features, particularly in scenarios with limited samples. To tackle these challenges, this paper presents a novel perspective on the AI-generated image detection task, advocating for detection under semantic-decoupling conditions. Building upon this insight, we propose SemGIR, a semantic-guided image regeneration based method for AI-generated image detection. SemGIR first regenerates images through image-to-text followed by a text-to-image generation process, subsequently utilizing these re-generated image pairs to derive discriminative features. This regeneration process effectively decouples semantic features organically, allowing the detection process to concentrate more on the inherent characteristics of the generative model. Such an efficient detection scheme can also be effectively applied to attribution. Experimental findings demonstrate that in realistic scenarios with limited samples, SemGIR achieves an average detection accuracy 15.76% higher than state-of-the-art (SOTA) methods. Furthermore, in attribution experiments on the SDv2.1 model, SemGIR attains an accuracy exceeding 98%, affirming the effectiveness and practical utility of the proposed method.

## CCS CONCEPTS

• **Security and privacy → Human and societal aspects of security and privacy**.

## KEYWORDS

AI-generated Image Detection, Semantic Image Regeneration

## 1 INTRODUCTION

The rise of image generative models, including VAE [16], GAN [8], Diffusion [25] and their variations [2, 11, 13, 14, 25], has made

*ACM MM, 2024, Melbourne, Australia*
© 2024 Copyright held by the owner/author(s). Publication rights licensed to ACM.
ACM ISBN 978-x-xxxx-xxxx-x/YY/MM
https://doi.org/10.1145/nnnnnnn.nnnnnnn

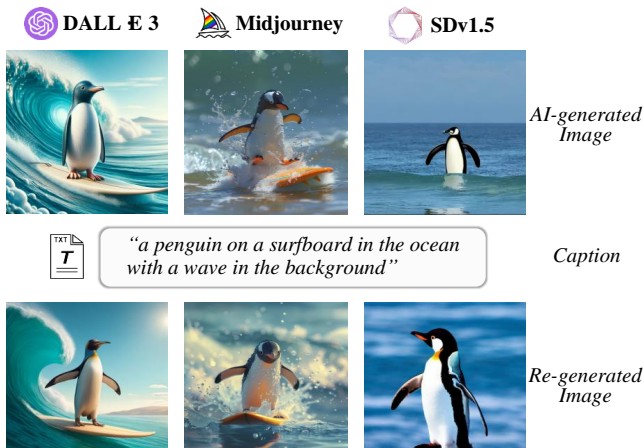

**Figure 1: Images re-generated using the same generative model as the AI-generated image demonstrate consistency.**

it possible for people to create images corresponding to simple text descriptions. However, this has also raised concerns. Some unscrupulous individuals may abuse these technologies, using image generative models to create false information, spread rumors, and even manipulate public opinion. For example, an image of "explosion at the Pentagon" on social media caused a stir in the U.S. stock market [4]. However, it is an AI-generated image rather than not a real image. The fact that merely an AI-generated image can cause such a significant upheaval fully demonstrates the harm of misusing image generation technology. Developing effective detectors for AI-generated images is crucial.

Existing AI-generated image detection techniques predominantly employ a learning-based paradigm, wherein discriminative features are extracted from images and subsequently utilized to train classification models [1, 6, 9, 12, 18, 20, 29, 31, 32]. Earlier methods extract features by directly employing convolutional neural networks (CNNs) from images [1, 29, 31]. UnivFD [20] extracts discriminate features using a large pre-trained vision-language model and then employs a nearest neighbor classifier to detect AI-generated images.

In our view, *the generation process can be seen as a couple of text prompts and inherent characteristics of the generative model.* To achieve a more general detection, the discriminative features shall rely more on the inherent characteristics features of the model instead of text prompts. Therefore, these learning-based detection methods often need large-scale datasets generated with various different prompts, so as to eliminate the interference of text prompts and then focus on the inherent characteristics of the generative model, thereby achieving good generalization ability and high accuracy.

However, collecting data is time-consuming and laborious. This challenge is exacerbated when it comes to closed-source commercial generative models, as acquiring extensive training datasets can be prohibitively expensive and, at times, unfeasible due to intellectual property protection measures. Such a challenge naturally leads to a requirement: achieving effective detection in scenarios with limited samples. Unfortunately, directly applying the previous detection mechanisms in this scenario will result in subpar performance. Therefore, in this paper, our goal is to design a detection method that remains effective in scenarios with limited samples.

In order to achieve this goal, our first step will involve analyzing the inefficacy of previous detection mechanisms. Our analysis reveals that the interference caused by text prompts is a key factor, as the restricted range of text prompts will introduce biased features during training, consequently impacting the generalizability of the detectors.

Building on this analysis, we provide a novel insight regarding scenarios with limited samples, that is *detecting should be conducted under the semantic-decoupled conditions.* DIRE [32] proposed an effective reconstruction based method to establish semantic-decoupled features, where they regenerate an image with a DDIM inversion followed by a diffusion process, and then learn discriminative features from the difference between the original image and the reconstructed image. However, such a method requires the generative process to be invertible, when faced with an irreversible generative process (e.g., GAN), such an approach cannot be used.

To address such limitation, we propose a more general framework **SemGIR**, which is based on a **Sem**antic-**G**uided **I**mage **R**egeneration method. Specifically, given a candidate image, we utilize image-to-text generation methods to extract text from the candidate image. Then, based on this text, we perform semantic image regeneration by a text-to-image generation model [25] so that the re-generated image is consistent with the candidate image in semantic content. Finally, we measure the similarity between the candidate and the re-generated images. SemGIR compels the detector to ignore the content of the images and focus solely on the inherent characteristic of the model expressed within them. Since the characteristic features expressed in the images are distinct, the detection method based on SemGIR does not require a large number of samples for learning to achieve a high detection accuracy.

Besides, AI-generated image attribution, determining which generative model generated the image, is also crucial for safeguarding image security, which offers a more detailed form of detection. Hence, it is essential to enhance the focus of the detector on the inherent characteristic of the model. In pursuit of this goal, we additionally propose a semantic condition classification method, named **SemGIR-A**, using the text extracted from the candidate image as a semantic condition to measure the similarity of candidate images and corresponding re-generated images.

Our main contributions can be summarized in the following four aspects:

- We have reconsidered the shortcomings of the current detection methods in dealing with the diversified generative model in scenarios with limited samples; that is, the text prompt in the generated image is coupled with the inherent characteristic of the generative model, which is difficult to

detect. Therefore, a semantic image regeneration method is proposed to strip the semantic content from the image and realize the organic decoupling of the two, making the detector focus on the inherent characteristics of the generative model.

- In view of the particularity of detection and attribution tasks, a corresponding detection method is proposed on how to use the regenerated images. For detection, we directly measure the similarity between images for detection. For attribution, a semantic conditional similarity measurement is proposed, to further mitigate the impact of the diverse text prompt.

- In experiments conducted on various generative models using only 2000 training images, SemGIR enhances the average detection accuracy by 15.76% compared to state-of-the-art (SOTA) techniques. Regarding attribution, SemGIR-A attains accuracies of 98.01% in tracing to the SDv2.1 models. These findings underscore the efficacy of our proposed method.

## 2 RELATED WORK

### 2.1 Image Generative Model

A generative model refers to a probabilistic model capable of randomly generating observable data. As images are one of the most commonly encountered data types, research on generative models for image generation is also widely explored. From Generative Adversarial Networks (GAN) [8] to diffusion models [28], these generative models [2, 11, 13, 14, 25, 35] have been lauded for their ability to generate lifelike images. Particularly, Text-to-Image models now can synthesize high-resolution images that conform to complex text prompts [7, 17, 24–27, 33, 34], and allow for a wide range of image editing [22] and other downstream applications.

During the early stages, one of the most prominent generative models was the Generative Adversarial Network [8]. These models consist of a generator and a discriminator, which engage in an adversarial learning process. Through this process, the generator learns to produce images that resemble the distribution of real images. Since the introduction of GANs, researchers have proposed various improvements and variants [2, 3, 13, 14, 21, 36], continuously enhancing the quality and stability of the generated images. In particular, StyleGAN and its improved version, StyleGAN2 [15], have achieved fine-grained control over image style and attributes by incorporating style transfer techniques, resulting in the generation of highly realistic images.

However, GANs often face challenges such as mode collapse. As a result, recent researchers have shifted their focus to the diffusion model as a promising alternative. The diffusion model is inspired by nonequilibrium thermodynamics by iteratively adding Gaussian noise to an image, and then learning the reverse diffusion process to reconstruct the original image from the noise. To enhance the controllability of generated images, researchers have proposed a series of text-to-image diffusion models such as ADM [5], VQDM [10], Stable Diffusion [25], and Midjourney[1]. These models take a description i.e., a piece of text prompt and random noise as inputs, and then denoise the image under the guidance of the prompt so that the resulting image matches the prompt. Thanks to the incorporation of the text prompt, the generated images not only contain

---

[1]https://www.midjourney.com/

 

the inherent characteristic of the generative model but also the thematic content of the text. These two elements are closely coupled together, allowing the generative models to offer users a more enriched and personalized visual experience.

## 2.2 Detection of AI-generated images

With the rapid progress of generative models, the authenticity and origin of generated images have become increasingly difficult to discern. This has led to a growing demand for detection methods to identify and distinguish AI-generated images.

Existing methodss [6, 12, 18–20, 29, 31, 32] for detecting AI-generated images primarily rely on the differences in the image features. CNNSpot [31] involves training a CNN as a binary classifier to directly distinguish between real and AI-generated images. Although this method is intuitive, its effectiveness relies on the similarity between the architecture of the generative model and the CNN network. Therefore, researchers are exploring alternative classification features to enhance the detection effectiveness. Fusing [12] proposes combining patch and global information of images to train a classifier, and GramNet [19] incorporates global texture features into the ResNet structure to improve the robustness and generalizability of AI-generated image detection. Meanwhile, FreDect [6] discovers that AI-generated images exhibit similar artifacts in the frequency domain caused by upsampling operations, and proposes a binary classifier based on frequency domain features.

Later, researchers discovered that pre-trained models can serve as effective feature extractors. LNP [18] extracts spatial image noise patterns based on a pre-trained denoising model. It then distinguishes between real images and AI-generated images based on the frequency domain of the noise patterns. LGrad [29] employs a pre-trained model to convert an image to be detected into a gradient map and normalize it, and then trains a binary classification model on the gradient map to identify AI-generated images. DIRE [32] proposes using a pre-trained diffusion model to reconstruct the image to be detected and judging the authenticity of the image by comparing the differences between the original and reconstructed images. However, DIRE performs well only on images generated by diffusion models. Subsequently, researchers began to focus on the generalizability of detection methods across architectures. UnivFD [20] argues that to enhance the generalization ability of detectors and enable them to reasonably detect AI-generated images, i.e., to learn a balanced decision boundary, a suitable feature space is required. Therefore, they utilized a pre-trained CLIP:ViT model to extract the feature space.

## 3 METHOD

In this section, we first outline the motivation behind proposing **SemGIR**. We then provide a detailed description of our method, as illustrated in Figure 3, focusing on three aspects: semantic-guided regeneration, feature extraction, and classification. We design different feature extraction strategies for the detection and attribution tasks.

## 3.1 Motivation

Regeneration is an effective method for disentangling the inherent characteristics of the model and text prompt within an image.

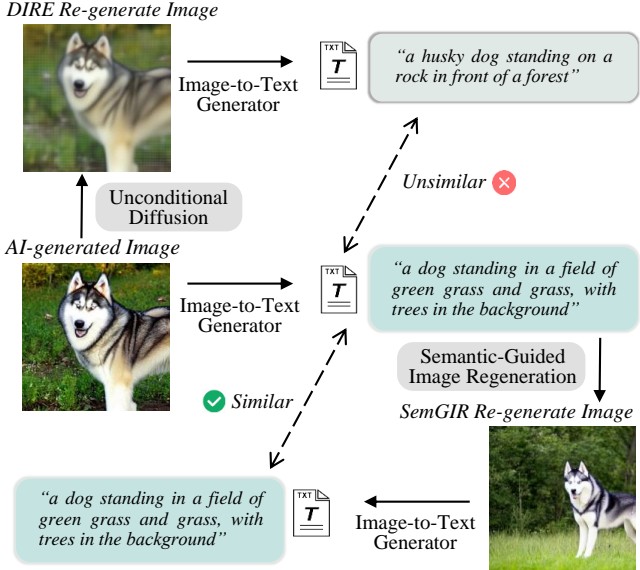

**Figure 2: The main difference between DIRE's reconstruction method and SemGIR's reconstruction method. DIRE is only a pixel-level reconstruction, while SemGIR is a semantic-level reconstruction that rebuilds the image using the text, ensuring the semantic consistency between the AI-generated image and the original image.**

DIRE [32] proposes a method that utilizes a diffusion model for pixel-level reconstruction of the image. The reconstruction error is then employed as a detection feature. From our perspective, this is a method of decoupling the text prompt and the inherent characteristics of the model. However, it can only identify images created through diffusion models. However, DIRE performs image reconstruction at the pixel level, which requires the presence of reversible structures in the generative model. Therefore, it is only applicable to stable diffusion models.

To achieve a general decoupling method against image generative models of different architectures, we propose to employ semantics to guide the regeneration of images, achieving semantic-level reconstruction. Compared to pixel-level reconstruction in DIRE, semantic reconstruction operates at a higher level of abstraction. Our proposed semantic reconstruction is independent of the model architecture and can be generalized to a wider range of generative models, addressing the generalization limitation of DIRE. Moreover, as semantic consistency is ensured between the original and regenerated images, it effectively removes the influence of text prompt, leading to a more thorough decoupling.

The examples shown in Figure 2 further validate our idea. The reconstructed image obtained using the reconstruction proposed by DIRE is only similar at the pixel level. Semantically, the two images differ significantly, so the influence of text prompt is not completely eliminated. On the other hand, using our proposed semantic-guided regeneration method for reconstruction ensures that the re-generated image and the AI-generated image are semantically consistent.

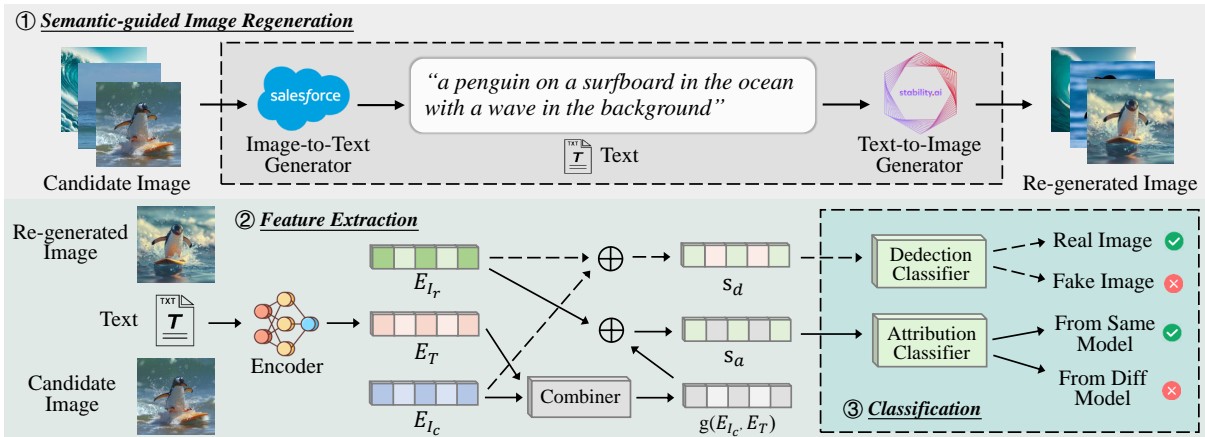

**Figure 3: An overview of our approach. Given a candidate image, we utilize image-to-text generator to extract text from the candidate image, and then perform semantic-guided image regeneration based on the text, ensuring that the re-generated image is consistent with the candidate image in terms of text content. For detection and attribution tasks, we design different feature extraction methods.**

## 3.2 Semantic-guided Image Regeneration

To achieve the decoupling of text prompt and the inherent characteristic of the model in images, we propose semantic-guided image regeneration. Given a candidate image $I_c$, we aim to obtain a semantic-guided regenerated image $I_r$. This process mainly involves two steps: image-to-text generation and semantic-guided image regeneration.

**Image-to-text Generation:** Given a candidate image $I_c$, we utilize a large pre-trained vision-language model $\mathcal{F}_1$ to get a corresponding text description $T$.

$$T = \mathcal{F}_1(I_c). \tag{1}$$

**Semantic-guided Image Regeneration** After obtaining $T$ of $I_c$, the next step is to utilize a pre-trained text-to-image generation model $\mathcal{F}_2$ to get $I_r$ that is semantically consistent with $I_c$ based on $T$. Specifically, given $T$, we use $\mathcal{F}_2$ to obtain $I_r$:

$$I_r = \mathcal{F}_2(T). \tag{2}$$

For different tasks, we adopt different model selection strategies.

For the detection task, since the generative model of $I_c$ is unknown, we uniformly use Stable Diffusion v1.5 (SDv1.5)[2] as the regenerative model. In fact, choosing different generation models has marginal impact on the detection performance, because the generation is only to inject semantic information to facilitate elimination. The results of the ablation experiment on regeneration model in Section 4.4, has also verified the inference.

For the attribution task. Given a set of models to be attributed $\{m_1, m_2, ..., m_k\}$, we use each model $m_i$, $i = 1, 2, ..., k$ in the set to regenerate $I_c$ and obtain the corresponding re-generated images $I_r^{m_1}, I_r^{m_2}, ..., I_r^{m_k}$.

$$I_r^{m_i} = m_i(T), \tag{3}$$

where $m_i$ belongs to $\mathcal{F}_2$.

The reason for using all the models to be attributed to regenerate the candidate image is that we believe the same model contains the

same inherent characteristics, while different models have different inherent characteristics. Therefore, we use all the generative models to be attributed to regenerate the candidate image and compare the similarity between these re-generated images and the candidate image. The generative model corresponding to the the re-generated image most similar to the candidate image is the generative model of the candidate image.

## 3.3 Feature Extraction

Building on previous foundations, we aim to select different feature extraction strategies for detection and attribution tasks, enhancing the decoupling effect. In this section, we will first introduce feature extraction for detection, followed by the introduction of semantic conditional feature extraction strategies for attribution.

**Feature Extraction for Detection.** We learn a separate encoder $\Phi(I)$ for images. With $\Phi(I)$, we encode the $I_c$ and $I_r$ to obtain their respective feature representations $E_{I_c}$ and $E_{I_r}$. Notably, we use the pre-trained model CLIP [23] to initialize the encoder $\Phi(I)$ for images. Thereafter, we concatenate $E_{I_c}$ with $E_{I_r}$ and get the embedding **s** as the similarity feature for detection task.

$$\mathbf{s}_d = E_{I_c}||E_{I_r}. \tag{4}$$

**Feature Extraction for Attribution.** In the detection task, we use a large pre-trained language model [23] to obtain features of $I_c$ and $I_r$. Then, directly concatenate them to serve as classification features. For the traceability task, we tried the same method but found it did not achieve the desired effect. We believe this is because the features extracted through the pre-trained model are not tightly bound to a particular generative model, making it difficult for the detector to learn the unique inherent characteristic of different generative models without a large number of samples. This suggests that for attribution tasks, more stringent constraints are needed, forcing the classification model to focus solely on the inherent characteristic of the generative model in the images. Based on this analysis, we propose a **semantic conditional classification** for

---

[2]https://huggingface.co/runwayml/stable-diffusion-v1-5

traceability tasks, using $T$ as a semantic condition for classification. The advantage of this approach is that it strictly limits the classifier to disregard the text prompt in the images, classifying only based on the inherent characteristic of the model, which helps to extract features bound to the generative model in AI-generated images.

Given $I_c$ and $I_r$, and $T$, we define the semantic conditional similarity feature $s$ as the similarity feature between $I_c$ and $I_r$ under the condition of $T$.

When extracting semantic conditional similarity features, the process is divided into three steps: feature extraction, feature fusion and feature concatenation.

(1) Feature Extraction: We learn separate encoders $\Phi(I)$ and $\Psi(T)$ for images and text. The $\Phi(I)$ is used to encode $I_c$ and $I_r$, obtaining their respective feature representations $E_{I_c}$ and $E_{I_r}$. Meanwhile, the $\Psi(T)$ is used to encode $T$, obtaining the text feature $E_T$. We also use the pre-trained cross-modal model CLIP [23] to initialize $\Phi(\cdot)$ and $\Psi(\cdot)$. The visual and text embeddings are aligned, making it easier to learn the combination between the $E_{I_c}$ and $E_T$.

(2) Feature Fusion: Following the approach of genecis [30], we use a function $\mathscr{G}$ to combine $E_{I_c}$ with $E_T$, obtaining the fused feature $g(E_{I_c}, E_T) \in R^N$.

$$g(E_{I_c}, E_T) = \mathscr{G}(E_{I_c}, E_T). \tag{5}$$

(3) Feature Concatenation: Finally, we use the feature obtained by concatenating $g(E_{I_c}, E_T)$ with $E_{I_r}$ as $\mathbf{s}_a$.

$$\mathbf{s}_a = g(E_{I_c}, E_T) || E_{I_r}. \tag{6}$$

## 3.4 Classification

Once the similarity feature $\mathbf{s_d}$ and $\mathbf{s_a}$ is obtained, the next step is to train a classification model. This model can be utilized to detect and attribute the generated images. For the detection task, real images are defined as the positive class, and AI-generated images as the negative class. For the attribution task, the images generated by the generative model to be attributed are defined as the positive class, and images generated by other models are defined as the negative class. Both detection and attribution tasks will use the same discriminator architecture and loss function.

We trained an end-to-end classification model to directly learn how to classify based on the extracted features. The classification model uses fully connected layers and applies the ReLU activation function. During the training process, we utilize the cross-entropy loss function to optimize the parameters of the classifier. By minimizing the cross-entropy loss, the classifier learns how to complete the classification task based on the feature vectors.

$$\mathcal{L} = -[y \log(f(\mathbf{s})) + (1 - y) \log(1 - f(\mathbf{s}))]. \tag{7}$$

Where $f$ represents the classification model, $\mathbf{s}$ represents the similarity features, $\mathbf{s}_d$ for the detection task, $\mathbf{s}_a$ for the attribution task, and $y$ represents the true label.

## 4 EXPERIMENT

### 4.1 Implementation Details

**Datasets.** To comprehensively evaluate our method, we employed 6 mainstream models, including generative models based on diffusion models and GANs to generate the fake images, which are SDv1.5[3], SDv2.1[4], Midjourney[5], ADM [5], VQDM [10], and StyleGAN2. Among them, the images for the StyleGAN2 [15] model are contributed by CNNSpot [31], while the others are sourced from the DiffusionForensics dataset [32]. For the training set, we randomly selected 1000 images generated with each model and combined with 1000 real images. Such amounts align with the few-shot settings. As for the testing, 2000 images generated with each model are selected to evaluate the performance. Besdies, we also evaluate the JPEG robustness of the detection, which is done by testing the detection/attribution accuracy on the JPEG compressed images.

**Evaluation Metrics.** Both the detection and attribution task of the AI-generated image are regarded as a binary classification task to be evaluated. Detection aims to correctly distinguish between real images and AI-generated images. Attribution aims to detect the images generated by specifc models. Both the accuracy can be defnded as following manner:

$$ACC = (N_+ + N_-)/N_{total}$$

In the detection task, $N_+$ indicates the number of the real images that are correctly detected as real and $N_-$ denotes the number of the fake images that are correctly detected as fake. $N_{total}$ represents the total number of testing images. For the attribution task, when attribute specific model $\mathcal{M}_a$, all the test images are divided into two classes: the images generated with $\mathcal{M}_a$, denoted as $I_{\mathcal{M}_a}$ and the other images that are not generated with $\mathcal{M}_a$, denoted as $I_{\mathcal{M}_o}$. $N_+$ indicates the number of the images in $I_{\mathcal{M}_a}$ whose source model is correctly attributed, $N_-$ illustrates the number of the images in $I_{\mathcal{M}_o}$ which are not attributed to $\mathcal{M}_a$. We adopt accuracy and average precision in our experiments. Due to space limitations, the average precision results are provided in the supplementary materials.

**Baselines.** We select CNNSpot [31], FreDect [6], Fusing [12], Gram-Net [19], LNP [18], LGrad [29], DIRE [32], and UnivFD [20] as our baseline methods. To ensure a fair comparison, all baselines are retrained on our few-shot training set. For methods that require pre-trained models for feature extraction, including DIRE, LNP, and LGrad, we follow the their settings and use the officially provided pre-trained models for image reconstruction[6], gradient map acquisition, and noise extraction, respectively.

**Training Details.** We initialize the backbone of the image and text feature extractors and the combiner network using the ResNet50×4 CLIP model. For the Image-to-Text model, we chose the BLIP model. For the classifier, we design a binary classification network consisting of two fully connected layers. The first layer accepts input features of $640 \times 2$ dimensions and outputs features with 640 dimensions. The second layer performs binary classification based on the 640-dimensional features. Among the architecture, "ReLU" is utilized as the activation function. We set the batch size as 32, the learning rate as 1e-4 and adopt Adam as the optimizer. Training is performed on an RTX 4090 GPU.

### 4.2 Detection Effectiveness

The detection accuracy of SemGIR and other baselines is shown in Table 1. It can be seen that our method outperforms the baselines

---

[3]https://huggingface.co/runwayml/stable-diffusion-v1-5
[4]https://huggingface.co/stabilityai/stable-diffusion-2-1
[5]https://www.midjourney.com/
[6]https://openaipublic.blob.core.windows.net/diffusion/jul-2021/lsun_bedroom.pt

**Table 1: The detection accuracy comparison between SemGIR and baselines. Among all detectors, the best result and the second-best result are denoted in boldface and underlined, respectively.**

| Generator | Detection Methods | | | | | | | | |
|---|---|---|---|---|---|---|---|---|---|
| | CNNSpot | FreDect | Fusing | GramNet | LGrad | LNP | DIRE | UnivFD | SemGIR |
| SDv1.5 | 0.5630 | 0.5250 | 0.9960 | 0.7950 | 0.8730 | **0.9980** | 0.7980 | 0.8840 | 0.9929 |
| SDv2.1 | 0.7937 | 0.4850 | 0.9405 | 0.6680 | 0.8710 | **0.9985** | 0.9710 | 0.7960 | 0.9561 |
| Midjourney | 0.5763 | 0.5145 | 0.7463 | 0.8136 | 0.9845 | **1.0000** | 0.9522 | 0.8327 | 0.9954 |
| ADM | 0.6735 | 0.5250 | 0.4990 | 0.4995 | 0.8205 | 0.4880 | 0.5130 | 0.6380 | **0.9960** |
| VQDM | 0.4980 | 0.5005 | 0.5220 | 0.6455 | 0.8725 | 0.7665 | 0.6320 | 0.8285 | **0.9970** |
| StyleGAN2 | 0.6260 | 0.5080 | 0.9370 | 0.7235 | 0.5250 | 0.6345 | 0.5147 | 0.8970 | **0.9546** |
| Average | 0.6217 | 0.5096 | 0.7734 | 0.6908 | 0.8244 | 0.8142 | 0.7301 | 0.8127 | **0.9820** |

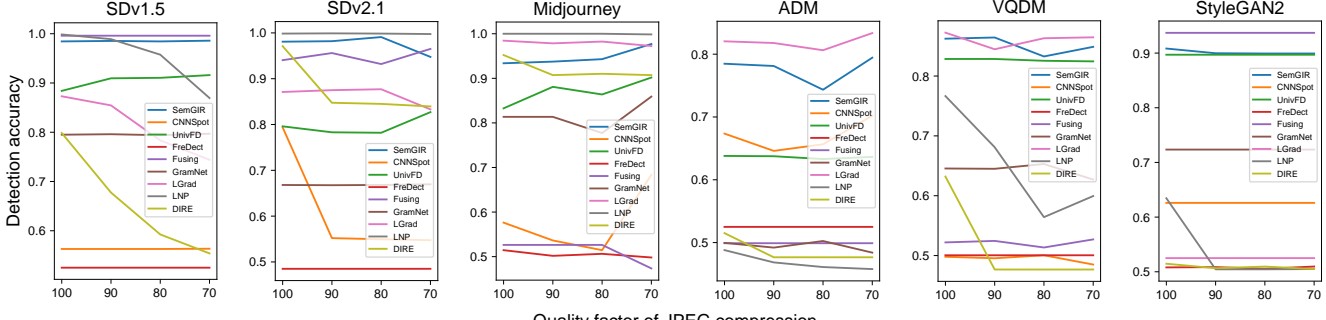

**Figure 4: The accuracy change of SemGIR and baselines under different qualities of JPEG compression.**

in terms of average detection accuracy, with a clear advantage. Our method achieves an average accuracy of 98.2% on six generative models, while the highest average accuracy of the baseline methods is only 82.44%. This indicates that our proposed method performs superior in scenarios with limited samples.

However, it is also observed that LNP behaves slightly better regarding the images generated with SDv1.5, SDv2.1 and Midjourney, but the generalizability to other models exhibits a clear drop. Such a generalizability issue can be found in all the compared methods. However, with the proposed SemGIR, all the accuracy stays at a high level, which is higher than 95%. Such a positive results indicates the strong generalizability of SemGIR. We conclude the reason as: SemGIR separates the intrinsic feature of the generative model from the textual content in the image. By reducing the influence of various prompts, the detector can concentrate on the distinguishing feature of the generative model. Consequently, a small number of samples are adequate for training the classifier effectively.

For DIRE, which is also a re-generation-based mechanism, the detection of the invertible structure (diffusion model) is much higher than that of GAN, such results align with our analysis. Compared with DIRE, SemGIR outperforms a lot, which greatly indicates the importance of semantic-conditioned detection and the semantic level reconstruction proposed in this paper.

**Robustness Test**: In real-world applications, images propagated on online social networks may undergo various common image processing techniques, such as JPEG compression. Therefore, it

is crucial to evaluate the performance of detectors in handling distorted images. We apply JPEG compression at different quality factors to test the performance of each detector under a lossy environment. Figure 4 illustrates the accuracy change of SemGIR and baselines in this scenario.

It can be seen that our proposed method exhibits robustness across nearly all models. Even with compressed images, the detection accuracy is still at a high level. For Stable diffusion, the most commonly used generation model, the detection accuracy is higher than 93%, which greatly illustrate the robustness of the detection. We believe the reason comes from the invariance of the semantic features of the images. Even under different level of compression, the BLIP model still can extract same semantic feature for re-generation, such an invariance in semantics leads to the alignment of the semantic conditions, where making the detection still works in a semantic-guided manner. Such positive results greatly show the effectiveness of the proposed SemGIR in practical scenarios.

## 4.3 Attribution Effectiveness

As aforementioned, SemGIR can also be utilized as model attribution since different generative models may result in different inherent characteristics. By detecting with semantic aligned conditions, the features with different model can be effectively differentiated. In this section, we focus on investigating the attribution of the SDv1.5

**Table 2: The traceability accuracy comparison between SemGIR and baselines. Among all detectors, the best result and the second-best result are denoted in boldface and underlined, respectively. QF means the quality factor of JPEG compression.**

| Generator | JPEG Compression | Attribution Methods | | | | | | | | |
|---|---|---|---|---|---|---|---|---|---|---|
| | | CNNSpot | FreDect | Fusing | GramNet | LGrad | LNP | DIRE | UnivFD | SemGIR-A |
| SDv1.5 | Original | 0.7599 | 0.7563 | 0.8674 | 0.8148 | 0.8133 | 0.7387 | **0.9114** | 0.7748 | 0.9016 |
| | QF=90 | 0.7599 | 0.7563 | 0.6695 | 0.8150 | 0.8282 | 0.7317 | 0.7514 | 0.8001 | **0.8969** |
| | QF=80 | 0.7602 | 0.7563 | 0.6608 | 0.8143 | 0.8011 | 0.7317 | 0.6850 | 0.7799 | **0.9088** |
| | QF=70 | 0.7609 | 0.7563 | 0.2514 | 0.8138 | 0.7828 | 0.7317 | 0.7061 | 0.7682 | **0.9266** |
| SDv2.1 | Original | 0.8021 | 0.8332 | 0.9718 | 0.8054 | 0.8433 | 0.9701 | 0.9020 | 0.8040 | **0.9801** |
| | QF=90 | 0.8007 | 0.83344 | 0.9268 | 0.8052 | 0.8066 | **0.9935** | 0.7514 | 0.8040 | 0.9781 |
| | QF=80 | 0.8013 | 0.8332 | 0.6503 | 0.8052 | 0.7674 | 0.9088 | 0.7360 | 0.8040 | **0.9597** |
| | QF=70 | 0.8009 | 0.8334 | 0.3752 | 0.8050 | 0.7584 | **0.9064** | 0.7134 | 0.8040 | 0.8935 |

and SDv2.1 models and test the robustness of various methods under JPEG attacks.

In this work, we focus on investigating the attribution of the SDv1.5 and SDv2.1 models and test the robustness of various methods under JPEG attacks. Table 2 presents the attribution accuracy and robustness of our method and other baselines. We also conducted more detailed attribution experiments and the results can be found in the supplementary material.

For the attribution of SDv1.5, the best and second-best results are achieved by our method and DIRE, respectively, both reaching over 90%. This indicates that extracting the the inherent characteristics of the model is a highly effective approach for the attribution task, as it allows obtaining features bound to the model. As for the attribution of SDv2.1, our method achieves the highest accuracy, which is 98.01%. Both experiments demonstrate the superiority of our proposed method in the attribution task.

We also find that attributing SDv1.5 is more challenging than attributing SDv2.1, with most methods achieving higher attribution accuracy on SDv2.1 compared to SDv1.5. We believe that the reason for the performance decline is that the inherent characteristics of SDv2.1 model are easily distinguishable from other models. In contrast, SDv1.5 may share similar intrinsic properties with other models.

**Robustness Test**: Under JPEG compression processing, some methods experience a decrease in performance. However, our method remains relatively stable, especially for the attribution experiment on SDv1.5, demonstrating the robustness of our method. This can still be attributed to the stability of the BLIP and CLIP models when facing attacks, enabling the SemGIR method to maintain robustness under attacks. At the same time, this is also due to the stability of the features selected by our method. We believe that the inherent characteristics of the model are more stable than semantic information when facing attacks. The stability in these two aspects makes our method more robust.

## 4.4 Ablation Studies

For the detection and attribution experiments, we respectively investigated the influence of different modules on the detection and attribution accuracy. Specifically, we investigated the impact of the regeneration process, the selection of regeneration models, and

the feature extraction module on the accuracy of detection and attribution. The experimental results are shown in Table 3.

**The Importance of Regeneration Process.** The core of our method lies in semantic-guided regeneration. By comparing the accuracy of methods with and without the regeneration process, we further emphasize the importance of the regeneration process. When the regeneration process is not employed, we can only obtain the candidate image. Alternatively, we can further perform caption extraction on the candidate image, obtaining both the candidate image and its corresponding caption. The results are presented in the "Without Regeneration Progress" section of Table 3.

When using only the candidate image as the classification feature, the average accuracy is merely 85.34%, especially for the ADM, VQDM, and StyleGAN2 models, where the accuracy is only 75%. This indicates that using only the candidate image as a feature makes it difficult to effectively distinguish between real images and AI-generated images.

Compared to using only candidate images, when using the joint features of candidate image and caption for detection, the detection accuracy is lower, with an average detection accuracy of only 83.53%. This further demonstrates that when distinguishing between real images and AI-generated images in scenarios with limited samples, text should not be introduced as a feature. We believe there are two reasons for this. The first is that in small sample scenarios, the content that the detector can learn is limited, so introducing text and image features will make the features complex and not conducive to detector learning. The second reason is that since the text content is encoded through CLIP, it also belongs to the generated content. Therefore, mixing generated content into real images will make the features of natural images impure, making it difficult to distinguish them from the features of generative model images.

Consistent with the ablation experiment for detection, we used the attribution accuracy in the cases of using only candidate images and text, and using only candidate images. It can be seen that when using only candidate images for attribution, the accuracy is only around 80%, because generated images often exhibit similar features, making it difficult to perform attribution. However, after introducing text features, the attribution accuracy reached 85.01% and 94.97%, respectively, indicating the positive impact of introducing text on attribution accuracy. It can provide more features to help

**Table 3: Ablation study of SemGIR. "without Regeneration Process" investigates the impact of not using the regeneration process on performance, "Regeneration Model" represents the influence of the selection of regeneration model during the regeneration process on performance, and "Feature Extraction" represents the impact of feature selection on performance.**

| Task | Generator | SemGIR | Without Regeneration Progress | | Regeneration Model (SDv2.1) | Feature Extraction |
|------|-----------|--------|------------------------|------------------|------------------------------|--------------------|
| | | | Candidate image-Caption | Candidate Image | | |
| Detection | SDv1.5 | **0.9929** | 0.9808 | 0.9456 | 0.9400 | 0.9844 |
| | SDv2.1 | 0.9561 | 0.9793 | 0.9400 | **0.9924** | 0.9808 |
| | Midjourney | **0.9954** | 0.8143 | 0.9458 | 0.9862 | 0.9338 |
| | ADM | **0.9960** | 0.7011 | 0.7339 | 0.9783 | 0.7848 |
| | VQDM | **0.9970** | 0.8760 | 0.7923 | 0.9748 | 0.8624 |
| | StyleGAN2 | 0.9546 | 0.6603 | 0.7631 | **0.9622** | 0.9083 |
| | Average | **0.9820** | 0.8353 | 0.8534 | 0.9723 | 0.9091 |
| Attribution | SDv1.5 | **0.9016** | 0.8501 | 0.7559 | 0.8767 | 0.8598 |
| | SDv2.1 | **0.9801** | 0.9497 | 0.8040 | 0.9351 | 0.9670 |

the detector with attribution, which also proves the correctness of our idea of introducing text-conditional similarity features for attribution. After using re-generated and text-conditional similarity features, the attribution accuracy for SDv1.5 and SDv2.1 reached 90.16% and 98.01%, respectively.

**The Impact of Regeneration Model.** In re-generation, the selection of the re-generation model is also a very important part. To investigate the impact of different re-generation models on the results, we used SDv2.1 as the regeneration models. The results displayed in the "Regeneration Model" of Table 3 show the outcomes of using the SDv2.1 model instead of the SDv1.5 model for regeneration.

From the table, it can be seen that using SDv1.5 and SDv2.1 as re-generation models does not have a significant impact on the results, with average accuracies of 98.20% and 97.23%, respectively. This proves our previous point that the similarity between AI-generated images and re-generated images is greater than the similarity between human-generated images and re-generated images. It is worth noting that for the SDv1.5 model, the accuracy of using the corresponding model as the regeneration model is higher than the accuracy of using SDv2.1 as the regeneration model, which are 99.29% and 94.00%, respectively. The same trend is also reflected in SDv2.1. Using the SDv2.1 model as the re-generation model achieves a higher detection accuracy than using SDv1.5, which are 95.61% and 99.24%, respectively. From this point of view, it also proves the rationality of our proposed attribution method, that is, the same model exhibits the same inherent characteristics of the model, while different models have different inherent characteristics of the model. When using models different from the source of the AI-generated image for detection, due to the different inherent characteristics of different models, the re-generated images show a slight decrease in similarity with the AI-generated images, leading to ambiguity in the decision boundary and thus a decrease in detection accuracy.

Based on the above analysis, When using a different model from the model to be attributed for re-generation, we can see a significant decrease in accuracy. When attributing SDv1.5, if SDv2.1 is used as the re-generation model, the accuracy decreases by 2.49%.

When attributing SDv2.1, the accuracy decreases by 4.5%. This also simultaneously verifies our previous viewpoint: the same model contains the same inherent characteristics ,while different models contain different inherent characteristics.

**The Impact of Feature Extraction.** Finally, we investigated the impact of the feature extraction module on the accuracy of detection and attribution. In the detection task, the similarity features between the candidate image and the re-generated image are extracted; while in the attribution task, the similarity features under semantic conditions are extracted. To demonstrate the influence of different feature extraction methods on accuracy, we extracted semantic conditional similarity features for the detection task and tested the accuracy, finding that the average accuracy was only 90.91%. This indicates that semantic conditional features should not be introduced in distinguishing between real images and generated images, because semantic conditional features belong to the generated part. When the features of real images are combined with semantic conditional features, the presence of generated components in the supposed real features can lead to misjudgments.

In the attribution task, we only use the features of the candidate image and the re-generated image for attribution. When using only image features for attribution of the two models, the accuracy is only 85.98% and 96.70%. This demonstrates the necessity of selecting semantically similarity features for the attribution task.

## 5 CONCLUSION

In this paper, we have presented a novel approach for detecting AI-generated images by employing semantic-guided image regeneration to mitigate the influence of text prompt.

Our method compares the similarity between the input image and its regenerated version using neural networks, enabling reliable detection. Furthermore, we have introduced a semantic conditional image similarity detection method to enhance the accuracy of image attribution by further eliminating text prompt interference. Our reconsideration of text prompt interference elimination opens up new avenues for AI-generated image detection research. This direction encourages the exploration of more efficient methods to tackle the challenges posed by text prompts.

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
