# OpenReview forum: "SemGIR: Semantic-Guided Image Regeneration based method for AI-generated Image Detection and Attribution"
_acmmm.org/ACMMM/2024/Conference — MM2024 Poster_

### Official Review · Reviewer_qZ7x · 2024-05-21

**Rating:** 4
**Confidence:** 2

**Summary:**

This article describes a method for detecting AI-generated images in a few-shot learning scenario. Specifically, the authors utilize a non-learned text-conditional image generation approach to create contrastive images for classification and attribution learning. The experiments validate the effectiveness of this method in few-shot learning scenarios.

**Strengths:**

Using text-conditional image generation as a method to produce contrastive images for classification and attribution learning sounds quite reasonable. This approach effectively utilizes the differences in images synthesized by different generation models with the same text content to learn the unique traces of each generation model.

The performance of this method under few-shot learning conditions is satisfactory.

**Limitations:**

The motivation of some details are not very clear, for instance, in Line 486, why are the features of the text and the Candidate Image concatenated, rather than the features of the text and the regenerated Image? Both the Candidate Image and the regenerated Image correspond to the same text context. This difference in treatment is confusing.

Since the experiments revolve around a few-shot learning environment with only 2,000 training images, the author should conduct ablation studies on the number of training samples to explore how the performance of the method and compared works changes with more or fewer training samples.

Furthermore, the initial discussion about the differences between this paper and DIRE is confusing. The core of DIRE involves using reconstruction to obtain residuals for AI-generated image detection, which is different from the motivation of this paper.
Also, the statement in Line 142 is inaccurate; GANs can be reversed through direct optimization techniques, as evidenced by research in the GAN inversion field. Clearly, there is no significant difference between diffusion models and GANs in terms of reversibility.

**Suitability:**

3

---

### Official Review · Reviewer_onWs · 2024-05-22

**Rating:** 3
**Confidence:** 4

**Summary:**

Focusing on the AI-generated image detection task, this paper presents a semantic-guided image regeneration-based method. The image2text model will be first applied on the candidate image to get the corresponding text description, through which a text2image model will be applied to get the regenerated image. Then the features of these two images are utilized for detection or attribution.

**Strengths:**

The paper is easy to follow and the writing is quite well. The flowchart and the main structure of the proposed method are explained clearly and the design about considering text prompts in the image detection process provides insight to future work.

**Limitations:**

The motivation of the proposed method is quite unclear. Is it to leverage the “inherent characteristics of the generative model” or “achieving effective detection in scenarios with limited samples”? The experimental results do not show supporting evidence regarding the claim of the motivation.

GAN model could also be invertible through GAN inversion.

The Semantic-guided Image Regeneration module could be regarded as data augmentation. Considering the limited amount of training data, how to choose the style and content of training images to ensure the generalization of the proposed model?

The comparing methods all need a large amount of data rather than limited data, which affects fairness.

The experimental results do not provide an analysis of the model characteristics and whether the model has been overfitting.

Clerical error: “Dedection Classifier” should be detection error.

**Suitability:**

3

---

### Official Review · Reviewer_Y6Qj · 2024-05-25

**Rating:** 4
**Confidence:** 3

**Summary:**

The authors argue that the generation process could be a combination of text prompts and inherent characteristics of generative models. To enable effective detection, detectors should focus on the latter one. Therefore, the authors propose to decouple them by employing semantics to guide the regeneration of images.

**Strengths:**

1.	Different from pixel-level reconstruction, this paper employs a semantic-level reconstruction to decouple the inherent characteristics of generative models and text prompts.
2.	The authors conduct extensive ablation to validate the effectiveness of their method.

**Limitations:**

1.	What’s the difference between the GAN inversion and the invertible characteristic in Line 141?
2.	In Figure 2, the authors claim that DIRE achieves pixel-level reconstruction while SemGIR fulfills a semantic one. It would be better to verify it through a quantitative comparison on a large number of samples instead of a qualitative example.

**Suitability:**

3

---

### Meta-Review · Area_Chair_NHTR · 2024-07-01

**Recommendation:** Accept (Poster)
**Confidence:** 5

**Metareview:**

The strength of this paper includes novel method and satisfactory experimental results, while the weakness of this paper is unclear motivation as well as insufficient experiment analysis.  In the rebuttal, the authors have partially addressed these concerns, and the final ratings for this paper are 3 borderline accepts. Therefore, the AC recommend to accept this paper, but also encourage the authors to further improve the paper for publish.

---

### Meta-Review · Senior_Area_Chairs · 2024-07-10

**Recommendation:** Accept (Poster)
**Confidence:** 4

**Metareview:**

This paper received mixed ratings initially. After rebuttal, all the reviewers tend to accept the paper. SAC and AC agree with reviewers and recommend acceptance of the paper.